# Hybrid Sol–Gel and Spark Plasma Sintering to Produce Perovskite-like SrTiO₃ Ceramics for Radioactive Waste Isolation

Anton A. Belov [1], Oleg O. Shichalin [1,*], Evgeniy K. Papynov [1], Igor Yu. Buravlev [1], Erhan S. Kolodeznikov [1], Olesya V. Kapustina [1], Semen A. Azon [1], Nikolay B. Kondrikov [1], Alexander N. Fedorets [1] and Ivan G. Tananaev [1,2]

1 Nuclear Technology Laboratory, Department of Nuclear Technology, Institute of High Technologies and Advanced Materials, Far Eastern Federal University, 10 Ajax Bay, Russky Island, 690922 Vladivostok, Russia; nefryty@gmail.com (A.A.B.)

2 Institute of Chemistry and Technology of Rare Elements and Mineral Raw Materials, Kola Science Center, Russian Academy of Sciences, Akademgorodok, 26a, 184209 Apatity, Russia

\* Correspondence: oleg_shich@mail.ru

**Abstract:** The paper presents a reliable technology combining sol–gel synthesis and spark plasma sintering (SPS) to obtain SrTiO₃ perovskite-type ceramics with excellent physicomechanical properties and hydrolytic stability for the long-term retention of radioactive strontium radionuclides. The Pechini sol–gel method was used to synthesize SrTiO₃ powder from $Sr(NO_3)_2$ and $TiCl_3$ (15%) precursors. Ceramic matrix samples were fabricated by SPS in the temperature range of 900–1200 °C. The perovskite structure of the synthesized initial SrTiO₃ powder was confirmed by X-ray diffraction and thermal analysis results. Scanning electron microscopy revealed agglomeration of the nanoparticles and a pronounced tendency for densification in the sintered compact with increasing sintering temperature. Chemical homogeneity of ceramics was confirmed by energy dispersive X-ray analysis. Physicochemical characteristic studies included density measurement results ($3.11$–$4.80$ g·cm$^{-3}$), dilatometric dependencies, Vickers microhardness (20–900 HV), and hydrolytic stability ($10^{-6}$–$10^{-7}$ g·cm$^{-2}$·day$^{-2}$), exceeding GOST R 50926-96 and ISO 6961:1982 requirements for solid-state matrices. Ceramic sintered at 1200 °C demonstrated the lowest strontium leaching rate of $10^{-7}$ g/cm²·day, optimal for radioactive waste (RAW) isolation. The proposed approach can be used to fabricate mineral-like forms suitable for RAW handling.

**Keywords:** ceramics; perovskite; radionuclides; radioactive waste management; sol–gel synthesis; SPS

## 1. Introduction

The present stage of nuclear industry development inevitably leads to increasing amount of produced radioactive waste (RAW). The optimal way of utilizing spent radionuclides remains undetermined, stemming from the complexity of developing a safe and environmentally sound system of long-term RAW storage. This issue arises from several factors. Chiefly, the high radioactivity of much waste renders it hazardous to the surrounding environment for extraordinarily protracted durations. Secondarily, technologies capable of reliably isolating radionuclides from the biosphere for sustained periods are lacking. Furthermore, establishing systems of enduring disposal or reprocessing of RAW is associated with considerable financial and technological costs. An important factor also includes potential ecological and social risks under any scenario for RAW handling, engendered by the possibility of compromising the integrity of storage containers' isolating barriers [1,2]. The aforementioned difficulties, as well as a plethora of diverse approaches to their resolution, have heretofore prevented a consensus from emerging within the scientific community regarding an optimal model of waste treatment.

Further fundamental research is necessary to resolve the issue of RAW treatment. Contemporary science delineates principal directions for development concerning RAW synthesis and compaction. Specifically, methods have been proposed and tested, and they are thus intensely studied for producing materials meeting established standards regarding composition and form. This approach entails preparing initial RAW containing various nuclides and their subsequent thermal consolidation into a dense matrix [3–5]. The methodology allows for the extraction of radionuclides from primary wastes and their incorporation into an inert structure to ensure long-term isolation. Similar studies aim to develop technologies enabling transformation of RAW into a stable form, meeting safety requirements for transport and enduring storage. This approach could become one of the promising avenues for addressing challenges surrounding handling hazardous RAW.

A wide range of materials have been developed for the effective immobilization of radionuclides, including SYNROC-type composites [6–8]. SYNROC (an acronym for "synthetic rock") is a specially designed material for containing radioactive waste. SYNROC constitutes a mineral composite synthesized to mimic natural high-temperature minerals. The basis of SYNROC comprises perovskites, titanates, and zirconias capable of absorbing and enduringly retaining radionuclides. The structure afforded these SYNROC materials confers important advantages. Primarily, they exhibit high chemical stability, allowing the matrices to avoid breakdown due to water or other agents. Secondarily, SYNROC materials can accommodate a broad spectrum of radioactive elements. Tertiarily, these materials demonstrate high resistance to high temperatures and radiation effects.

It bears noting that perovskites have seen extensive and active investigation as matrices for radionuclide isolation [9–11]. The presence of perovskite structures in natural minerals stable over wide geochemical conditions for millions of years indicates the high hydrothermal stability of such compounds. This bodes well for the long-term safety of radioactive waste storage. The enhanced strength and density properties of perovskites are also of interest. Perovskites demonstrate considerable promise as durable containment media, building upon their proven longevity under demanding subsurface conditions comparable to anticipated repository environments. Further research seeks to optimize SYNROC–perovskite formulations and processing techniques to fully realize this potential for effective immobilization of diverse radionuclides in a minimal-impact form.

Perovskites can be of the simple $ABO_3$ type or more complex compositions depending on the A and B cations [12]. Examples of A cations include $Ce^{4+}$, $Nd^{3+}$, $Sm^{4+}$, $La^{3+}$, $Yb^{3+}$, and $Gd^{3+}$, while B cations comprise $Al^{3+}$, $Cr^{3+}$, $Fe^{3+}$, and $Ga^{3+}$. The ideal perovskite ($SrTiO_3$) possesses a cubic lattice structure, though this is only stable for a limited number of compositions. Cation substitutions within the lattice distort the structure, lowering symmetry and giving rise to tetragonal, orthorhombic, and monoclinic variations. Optimization of perovskite formulations for immobilizing diverse radionuclides requires consideration of these structural changes resulting from alterations in stoichiometry and cation selection/concentration. Further studies are exploring composition–property relationships to delineate formulations exhibiting enhanced durability under specialized waste storage conditions.

Among the recognized synthetic methods (mechanical activation/sintering, solution deposition, high-temperature gas or liquid plasma synthesis, molecular conductor synthesis, solid-state synthesis, etc.), the sol–gel technique stands out as a particularly promising approach, owing to its versatility in producing both nanocomposites and matrix materials [13,14]. The applicability of this technology spans a wide area—from inorganic matrices of varied constitution [15–18] to hybrid composites [19] and inorganic thin film structures [20]. The solution-based nature of sol–gel processing confers advantages such as scalability, precise control of composition, and capacity for synthesizing tailored nanostructured forms. Further exploration of this flexible methodology could consequently yield perovskite and composite formulations optimized for the stable immobilization of diverse radionuclides.

One development of the sol–gel technique is the Pechini method. Notably, this approach is used to synthesize perovskite metal oxides [11,21–24]. The process involves preparing solutions of target metal salts in ethylene glycol using an excess of a chelating agent (citric, glycolic, or lactic acid, EDTA). Heating the solution to 150 °C causes esterification and formation of an internal polymeric gel network. The gel is then converted to a powder by heating to 300 °C to remove most of the organic component, followed by calcination from 500 to 900 °C to yield the target oxide [25]. Dispersing the initial substances within the polymeric matrix ensures homogeneity and prevents phase segregation in the final product. Hence, the Pechini method enables efficient synthesis of perovskite oxides with tailored properties.

The final stage in forming solidified RAW is obtaining a rigid ceramic matrix. Immobilized within the matrix, RAW has qualified radionuclide isolation and prevention of environmental migration. This substantially heightens RAW stability over the long term, decreases thermogenicity, and simplifies transportation and storage. Beyond enhancing radiation safety, RAW compaction into such matrices opens avenues for utilizing the resultant material as an ionizing radiation source (IRS) across diverse domains, including nuclear physics, medicine, industry, defense, energy, and more. Continued investigation of matrix formulation and processing optimization remains vital to fully realize these benefits. Advanced techniques may allow tailoring matrices for specialized applications while ensuring radionuclides remain inertly fixed under all anticipated usage and disposal scenarios. With prudent development, immobilized IRS technology could thus offer promising solutions to numerous applications and to the overriding RAW remediation issue.

It is important to note that in obtaining compact ceramic blocks, efforts should aim to minimize their resultant volume compared to the original waste volume, i.e., achieve maximum possible densification of the solid substance structure. A dense matrix structure precludes dispersal of radioactive particles into the environment, ensuring long-term stability and impermeability. This allows the matrix as a stable RAW form to fix radioactive materials without degradation over extended timeframes. Thus, the sintering process plays a uniquely decisive and critically important role in matrix fabrication. Continued optimization of sintering parameters such as temperature, atmosphere, heating/cooling profiles, and additive composition represents a key research focus. Advancing sintering techniques could drive further matrix densification for enhanced radionuclide containment and durability suited for the demands of long-term waste storage and disposal.

Spark plasma sintering (SPS) has gained widespread use for sintering powder composites [26–33]. This technique involves pulsed electrical current heating of a sample under applied pressure. SPS was previously successfully utilized for $SrTiO_3$ synthesis aimed at strontium immobilization [34–36]. In these works applying SPS, we (i) developed a novel sintering–reaction synthesis method for $SrTiO_3$ perovskite ceramics immobilizing Sr-90; (ii) studied phase transformation kinetics in the $SrCO_3$-$TiO_2$ system from 20 to 1000 °C; (iii) determined optimal parameters for synthesizing $SrTiO_3$ ceramic with a density of 4.49 g/cm$^3$, hardness of 6.2 GPa, strength of 279 MPa, and Sr leaching rate of $10^{-5}-10^{-6}$ g/(cm$^2$·day); (iv) developed a synthesis method for $SrTiO_3$ ceramic immobilizing Sr-90 with a density of 95.6%, hardness of 1010 HV, and strength of 283 MPa; (v) obtained biphasic $SrTiO_3$–$TiO_2$ ceramic exhibiting quantum effects, prospective for thermoelectrics. SPS thus demonstrates potential for further optimizing immobilization matrices through enhanced densification.

In the present study, SPS was coupled with the Pechini synthesis method to prepare $SrTiO_3$ perovskite ceramics. The research aims included synthesizing powder via a sol–gel route from inorganic metal salt precursors, followed by sintering the resultant powder via SPS at various temperatures. Determination of physicomechanical properties and hydrolytic stability was also performed. Comparison with prior works will enable evaluation of qualitative characteristics of materials derived through different methods, identifying a preferable synthesis technique. The results may prove useful for solidifying RAW and developing immobilization matrices, as well as provide a basis for IRS. Con-

tinued refinement of the integration between Pechini and SPS processing has potential to optimize microstructure control and properties attainment conducive to radionuclide confinement functions.

## 2. Materials and Methods

### 2.1. Materials

The precursors used for sample synthesis were Sigma-Aldrich (St. Louis, MI, USA) reagents: strontium nitrate ACS reagent, $\geq$99.0% $Sr(NO_3)_2$, titanium(III) chloride solution $\geq$ 15% $TiCl_3$, monoethylene glycol $((CH_2OH)_2)$, and citric acid $(HOC(CO_2H)(CH_2CO_2H)_2)$.

### 2.2. Sol–Gel Synthesis

The powder was synthesized via the Pechini method as follows:

A total of 5.76 g of strontium nitrate $(Sr(NO_3)_2)$ was dissolved in 20 mL of water. Then, 27.99 g of titanium(III) chloride solution $\geq$ 15% was added. The solution was heated to 90 °C and evaporated. After 30 min with stirring, 35.36 g of citric acid and 12.56 mL of monoethylene glycol were added. The resulting mixture was evaporated with stirring for 1 hour at 90 °C until a viscous gel formed. The gel was then heat treated in air for 2 h at 400 °C to remove the organic component, followed by calcination in air for 2 h at 800 °C to remove residual carbon (heating for 1 h).

### 2.3. Material Characterization Methods

The ceramic samples were prepared by SPS using a "Dr. Sinter·LAB$^{TM}$" SPS-515S unit at temperatures of 900, 1000, 1100, and 1200 °C. The starting raw material was $SrTiO_3$ powder synthesized by the above-described method. The pressing pressure was 21.5 MPa, with a heating rate of 50 °C·min$^{-1}$. After holding for 5 min, the samples were allowed to cool uncontrolled to room temperature over half an hour.

Strontium concentration in leachate solutions was determined using a Shimadzu EDX-7000 (Kyoto, Japan) atomic absorption spectrophotometer. Scanning electron microscopy (SEM) was performed on a CrossBeam 1540 XB "Carl Zeiss" (Jena, Germany) microscope equipped with a Bruker (Mannheim, Germany) energy-dispersive X-ray spectroscopy (EDX) add-on. X-ray diffraction (XRD) was carried out using a "COLIBRI" diffractometer (Moscow, Russia). Vickers microhardness (HV) was determined at 0.2 N load using HMV-G-FA-D microhardness tester from Shimadzu (Kyoto, Japan). Experimental density (ED) was measured by hydrostatic weighing on an Adventurer$^{TM}$ balance from OHAUS Corporation (Parsippany, NJ, USA). Relative density (RD) was calculated as the ratio of experimental density (ED) to theoretical density (TD). Thermogravimetric analysis curves were recorded on a DTG-60H thermogravimetric analyzer from Shimadzu using platinum crucibles under dry argon flow at a heating rate of 10 °C/min from 35 to 1300 °C.

Evaluation of hydrolytic stability of $SrTiO_3$ matrices. Hydrolytic stability was assessed by monitoring the strontium leaching rate during long-term contact (30 days) of the matrix with distilled water (pH 6.8) at 25 °C under static conditions according to GOST R 52126-2003 (international analog ANSI/ANS 16.1). In particular, a cylinder-shaped ceramics sample (diameter 15 mm, height 4 mm) was placed into 55 mL of distilled water, and after a certain time period (1–30 days), the concentration of *Sr* in the supernatant was measured, while the pellet was removed, washed with distilled water, and put into the fresh portion of distilled water. The leaching rates were calculated according to the following equation:

$$R_n^{Sr} = \frac{m_n^{Sr}}{M_0^{Sr} \times t_n \times S} \tag{1}$$

where $R_n^{Sr}$—Sr leaching rate (g/cm$^2$ day); $m_n^{Sr}$—Sr mass, leached for nth time interval, g; $M_0^{Sr}$—Sr mass concentration in the matrix, g/g; *S*—the sample's surface area, cm$^2$; $t_n$—duration of the nth time interval, days.

The calculation of the effective diffusion coefficient (*De*) was performed by mathematical transformations of the second Fick law according to the method described in the paper [37]:

$$\frac{\sum m}{M_0} = 2\left(\frac{D_e}{\pi}\right)^{\frac{1}{2}} \times \left(\frac{S}{V}\right) t^{\frac{1}{2}} + \alpha \qquad (2)$$

where *m*—strontium weight, mg; *t*—leaching time, s; $M_0$—initial cesium content in the sample, mg; $D_e$—effective diffusion coefficient, cm$^2$/s; *S*—the surface area of the sample, cm$^2$; *V*—a volume of sample, cm$^3$; *α*—parameter that takes into account the initial leaching of strontium, not related to diffusion (strontium leaches out at the initial contact of the leaching solution with the sample surface).

In the calculation, this equation was reduced to a linear form by introducing the coefficient *K*, which represents the tangent of the slope of the straight-line dependence of strontium leaching from the sample on the square root of the contact time of the material with the leaching agent:

$$K = 2\left(\frac{D_e}{\pi}\right)^{0.5} \times \left(\frac{S}{V}\right) \qquad (3)$$

The effective diffusion coefficient was calculated:

$$D_e = \frac{K^2 \times \pi}{4} \times \left(\frac{V}{S}\right)^2 \qquad (4)$$

The leaching index (*L*) was calculated as the decimal logarithm of the inverse diffusion value:

$$L = lg\frac{1}{D_e} \qquad (5)$$

Estimation of the dominant leaching mechanism based on the dependence of the decimal logarithm of the accumulated fraction of leached radionuclide ($B_t$, mg/m$^2$) on the decimal logarithm of the leaching time *t*, s:

$$lg(B_t) = \frac{1}{2}lgt + lg\left[U_{max}d\sqrt{\frac{D_e}{\pi}}\right] \qquad (6)$$

where is $U_{max}$—the maximum amount of leached radionuclide, mg/kg, and *d*—matrix density, kg/m$^3$.

The leaching depth of the matrix characterizes the destruction of the matrix.

Matrix when it is in aqueous medium and is calculated according to Equation (7):

$$L_t^i = \sum_{i}^{n}\left(W_n^i\frac{t_n}{d}\right) \qquad (7)$$

where $L_t^i$—the leaching depth of the matrix reached during the time interval $t_n$, cm, and *d*—density of the sample, g/cm$^3$.

## 3. Results and Discussion

### 3.1. Preparation of Starting Mixtures

The results of the investigation of the mixture obtained by the sol–gel method are presented in Figure 1.

XRD patterns of the synthesized $SrTiO_3$ powder are shown in Figure 1a. The diffraction peaks of $SrTiO_3$ can be indexed to the (100), (110), (111), (200), (210), (211), (220), and (310) planes of the cubic crystal structure (JCPDS card no. 065089). This result indicates that the crystalline phase of the $SrTiO_3$ nanopowder did not contain any impurities. The average crystallite size was estimated to be 22 nm using the Debye–Scherrer formula. Rietveld refinement analysis was performed with profile reliability factors $R_p$ = 8.42% and $R_{wp}$ = 11.57%. The refinement parameters were as follows: weighted sum of squares

(WSSR) = 15,074.9, degrees of freedom (DoF) = 13,612, sum of squares SSR = $1.27845 \times 10^6$, and determination coefficient R2 = 0.99042. This indicates a good correspondence between the refined results and the experimental data. SEM images (Figure 1b) show that the powder particles ranged in size from 10 to 100 μm and consisted of agglomerates of 20–30 nm nanoscale primary particles, consistent with the Debye–Scherrer equation.

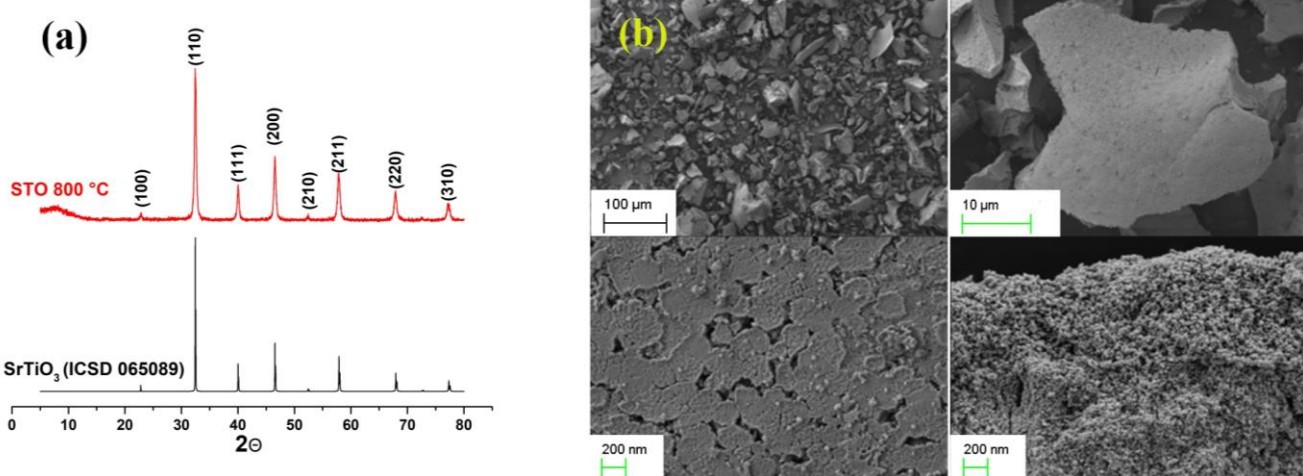

**Figure 1.** Characterization of the $SrTiO_3$ powder synthesized via the sol–gel method: (**a**) XRD patterns of the sample synthesized at 400 and 800 °C; (**b**) SEM images of the initial $SrTiO_3$ powder synthesized at 800 °C.

### 3.2. Characterization of Sintered Ceramics

3.2.1. Dilatometric Analysis

Dilatometric analysis indicated a single-stage powder compaction process (Figure 2). The onset of shrinkage was detected between 750 and 800 °C. Complete shrinkage was achieved at around 1100 °C, confirmed by the shrinkage curve plateauing. The single-stage behavior confirms the purity and homogeneity of the starting sol–gel powder.

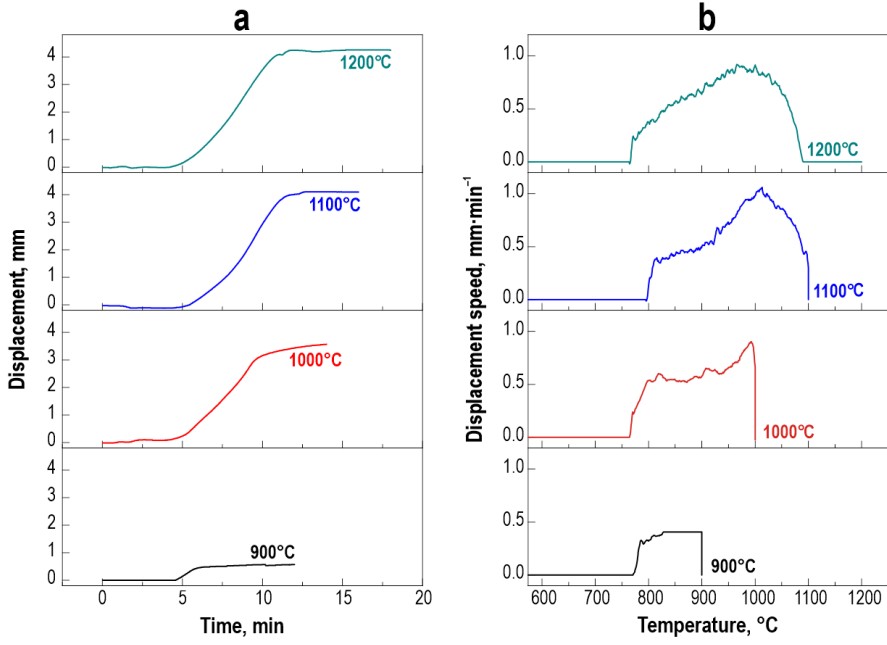

**Figure 2.** Densification kinetics during spark plasma sintering process: (**a**) dependency of displacement over time; (**b**) dependency of displacement rate over temperature.

This dilatometric curve differed from previous reports [27,28], likely due to the absence of chemical interaction between reactive components. The thermal compaction temperature range was also lower by 200–250 °C compared to previous studies. This is attributed to the high homogeneity and dispersion of the starting sol–gel SrTiO$_3$ powder.

### 3.2.2. Phase Composition and Structure

The XRD analysis results indicate that the ceramic obtained is composed of strontium titanate SrTiO$_3$ phase (ICSD 065089). Comparison with the XRD pattern of the sol–gel powder (Figure 3) shows no phase transformations or structural changes occurred. The intensity of the diffraction maxima increased with increasing sintering temperature. No additional phases were detected within the limits of XRD, indicating that the high-purity perovskite SrTiO$_3$ phase was formed after sintering. The increase in intensity with temperature suggests better crystallinity and grain growth at higher sintering temperatures.

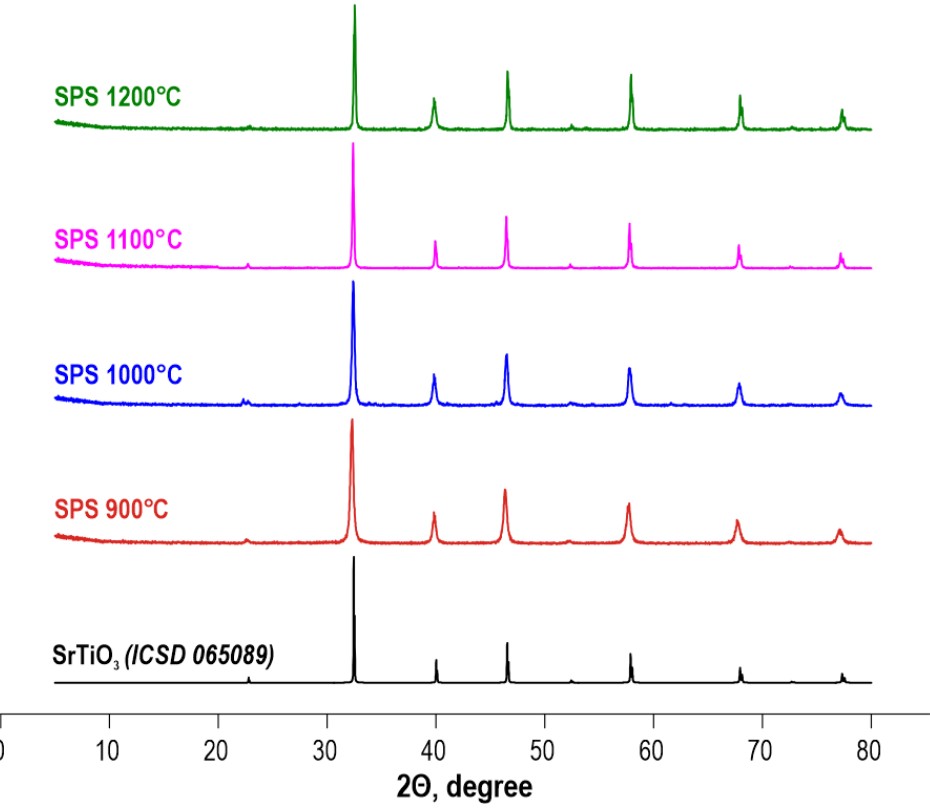

**Figure 3.** XRD patterns of the ceramic samples derived thereof at various SPS temperatures.

XRD analysis showed that increasing the temperature in the range of 900–1200 °C during sintering did not lead to changes in crystallographic composition. As seen in Figure 3a, three typical diffraction peaks of the perovskite structure were observed at 2θ = 22.758° (100), 32.391° (110), and 39.948° (111) in the sintered SrTiO$_3$ (STO) sample, indicating the presence of a cubic perovskite structure with space group Pm-3m, a = b = c = 3.904 Å, and 59.5 Å$^3$ volume. Moreover, only diffraction peaks of the STO phase were present in the STO sample, demonstrating the possibility of obtaining a single-phase SrTiO$_3$ round sample by sol–gel and cold pressing methods. The nanocrystalline SrTiO$_3$ sample after pressing at 21.5 MPa and sintering at 1200 °C for 5 min had a uniform perovskite structure that can be used for solidification of radioactive waste.

### 3.2.3. Morphology and Chemical Composition

The surface morphology and EDX analysis results of the sintered ceramics are presented in Figures 4 and 5.

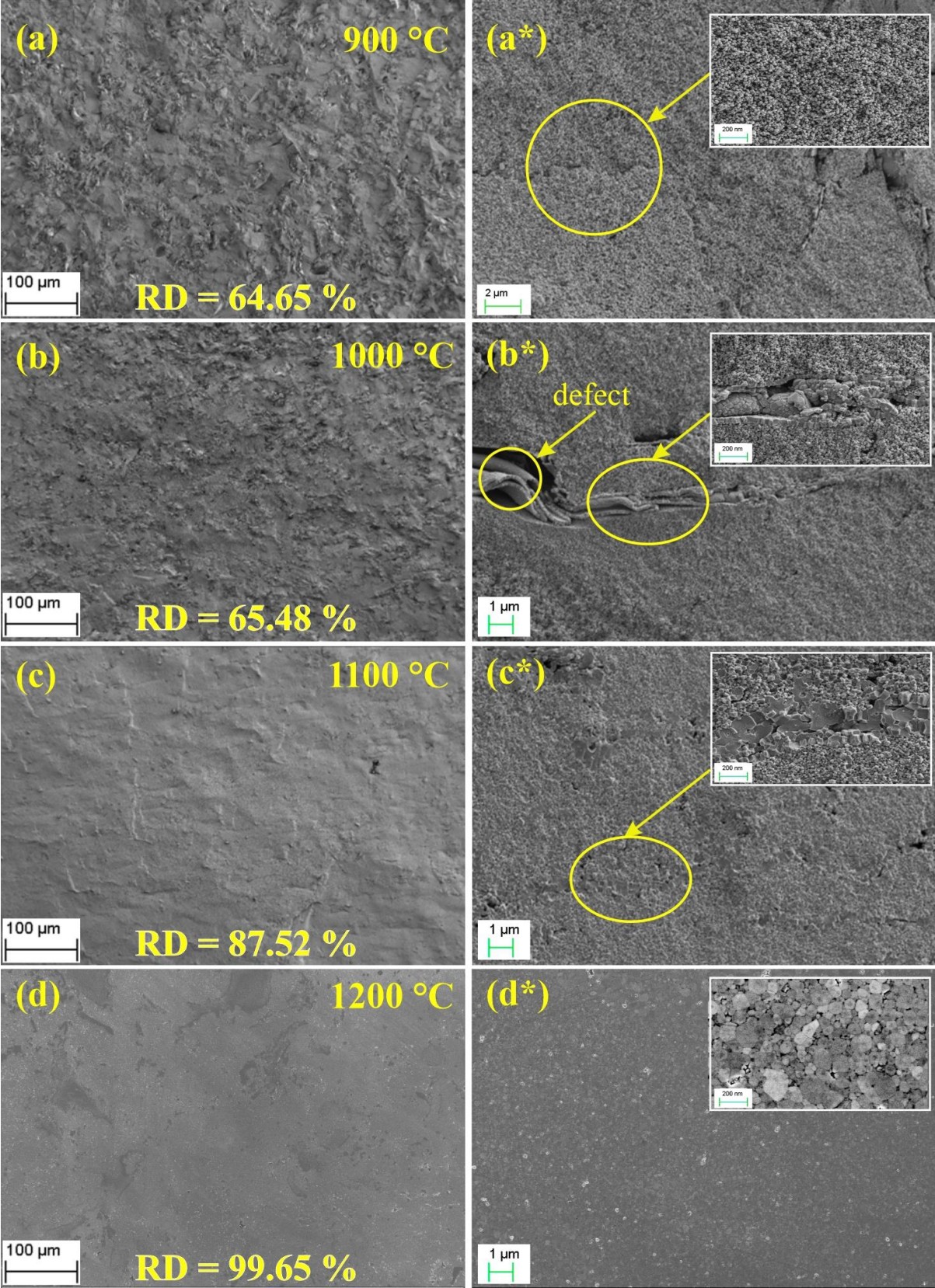

**Figure 4.** SEM images of SrTiO$_3$ samples obtained via SPS with temperatures: (**a**,**a***) 900 °C; (**b**,**b***) 1000 °C; (**c**,**c***) 1100 °C; (**d**,**d***) 1200 °C.

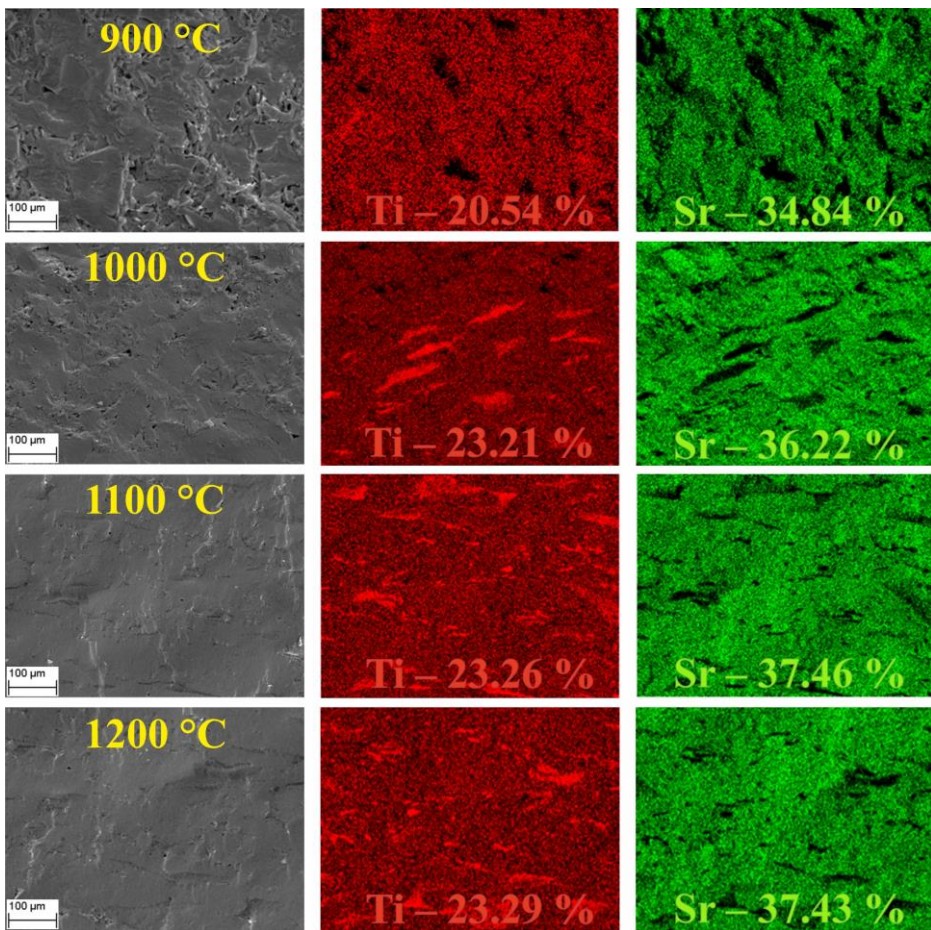

**Figure 5.** EDX images of sintering SrTiO$_3$ samples obtained via SPS with temperatures of 900, 1000, 1100, and 1200 °C.

According to the data obtained, the surface of the ceramics sintered at 900 °C consisted of agglomerated regions of nano-sized powder particles (Figure 4a,a*). The inset in Figure 4a* clearly shows a uniform distribution of nanoparticles of the original sol–gel powder. There were no noticeable areas of consolidation. Further increasing the temperature to 1000 and 1100 °C was accompanied by the formation of consolidated material regions (Figure 4b,b*,c,c*). The images of the sample (Figure 4b,b* *inset*) indicate the presence of a local overheating effect, since defects on the ceramic surface were clearly visible. Due to the relatively low heating rate, local formation of a monolithic structure is possible, leading to a change in the temperature gradient and formation of areas of different densities. The morphology of ceramics obtained at 1100 °C (Figure 4c,c* *inset*) showed that further increasing the temperature was accompanied by the disappearance of defective areas, but areas of non-monolithic structure were still present. SEM images of the sample obtained at 1200 °C indicated the formation of a monolithic structure. The presence of defects and unceramicized areas was not detected. The grains are clearly depicted in the inset in Figure 4b*. SEM data and dilatometry signaled the complete completion of thermal compaction and full ceramization of the sample. According to the EDX analysis results (Figure 5), an inhomogeneous distribution of titanium on the surface of the 1000 °C samples was noticeable, potentially serving as centers for the formation of areas with defects. EDX images of 1100 °C also indicated a decrease in defective areas and a tendency towards homogenization. However, areas of titanium concentration were preserved up to a temperature of 1200 °C. Comparison with previously obtained data indicated a similar ceramic formation mechanism involving partial agglomeration of areas followed by their growth. The distribution of titanium was also described by local concentrates in areas.

### 3.2.4. Mechanical Properties of the Samples

The results of the microhardness and density determination are presented in Figure 6. The ceramic underwent a significant increase in parameters with increasing temperature above 1000 °C. This behavior was due to the transition of the ceramic structure to a monolithic structure, which was confirmed by SEM analysis (Figure 4). The homogeneous distribution of elements and the absence of large defects also had a positive effect on the growth of physical parameters of ceramics. The obtained ceramics showed microhardness values lower by about 100 HV; however, they reached RD = 99.79% compared to reaction-bonded ceramics.

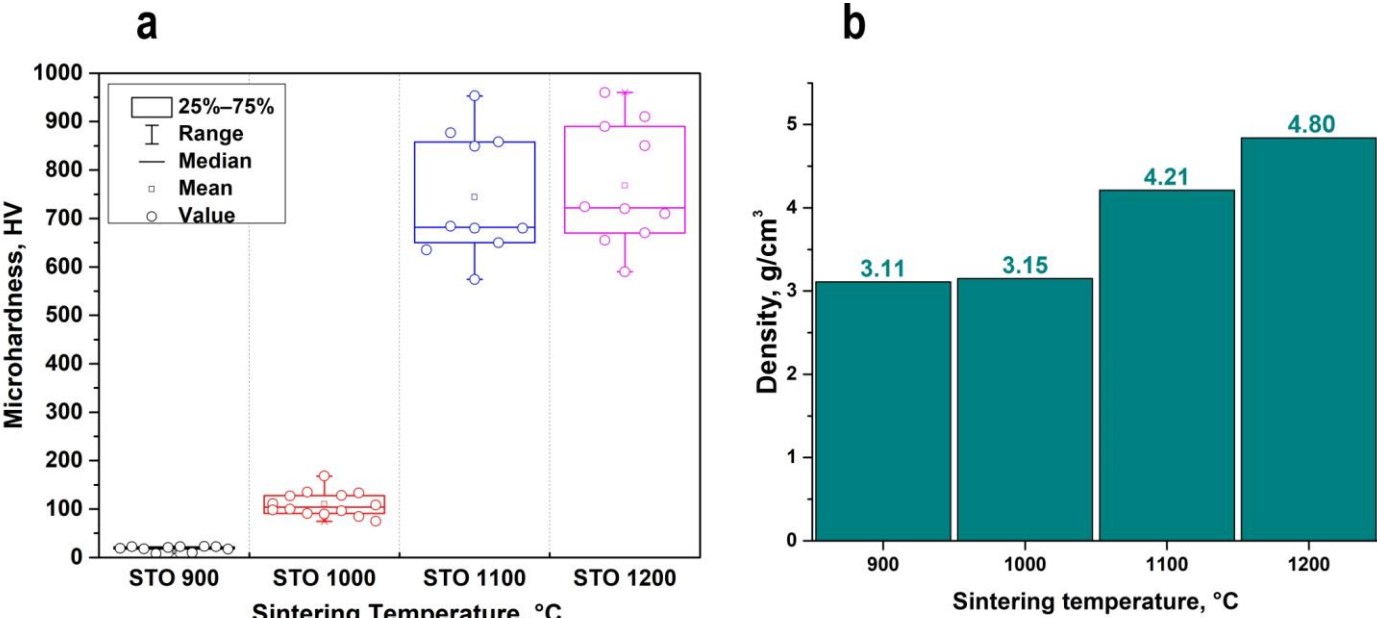

**Figure 6.** Physical and mechanical properties of the SrTiO$_3$ samples obtained via SPS with temperatures of 900, 1000, 1100, and 1200 °C: (**a**) dispersion of microhardness values; (**b**) density of the samples.

### 3.2.5. Hydrolytic Stability of the Samples

Some key physical factors influencing leaching include particle size, as leaching is partially dependent on the exposed surface area undergoing leaching. Homogeneity or heterogeneity of the solid matrix in terms of crystalline phases is also important. The time interval of interest and flow rate of the leaching fluid can impact results. Temperature during leaching, porosity of the solid matrix, and geometric shape and size of materials where diffusion processes dominate leaching kinetics should be considered. Permeability of the matrix during testing or under field conditions, as well as hydrogeological settings, may affect outcomes. Chemical factors commonly impacting leaching involve whether equilibrium or kinetic control governs release, the potential leachability of components, material or environmental pH (e.g., CO$_2$ influence), and complexation with inorganic or organic compounds.

The leaching resistance of SrTiO$_3$ ceramic matrices were evaluated to assess the composite materials obtained, as this is a key indicator of matrices effectiveness for strontium radionuclide immobilization. The results are shown in Figure 7a. The lowest leaching rate corresponded to the sample obtained at 1200 °C, which was $10^{-6}$–$10^{-7}$ g/cm$^2$·day, meeting the requirements of GOST R 50926-96 for solidified high-level waste. These high values were achieved firstly by the high degree of strontium chemical bonding within the source material structure. Secondly, the ceramization process was accompanied by a structural change in the material, forming a monolithic sample with the fewest defects and pores, preventing strontium leaching throughout the ceramic volume. Further calculations

of the tangent of the slope of straight lines showing the cumulative leach fraction versus the decimal logarithm of the leaching time (in seconds) (Figure 7b) differed significantly. However, it seemed possible to distinguish two patterns:

- the tangent of the slope of the straight line for the 900 °C sintered sample remained unchanged and was equal to 0.74, indicating strontium release was primarily due to the dissolution of the sample surface;
- the tangents of the slope of the 1000, 1100, and 1200 °C sintered samples significantly increased during leaching.

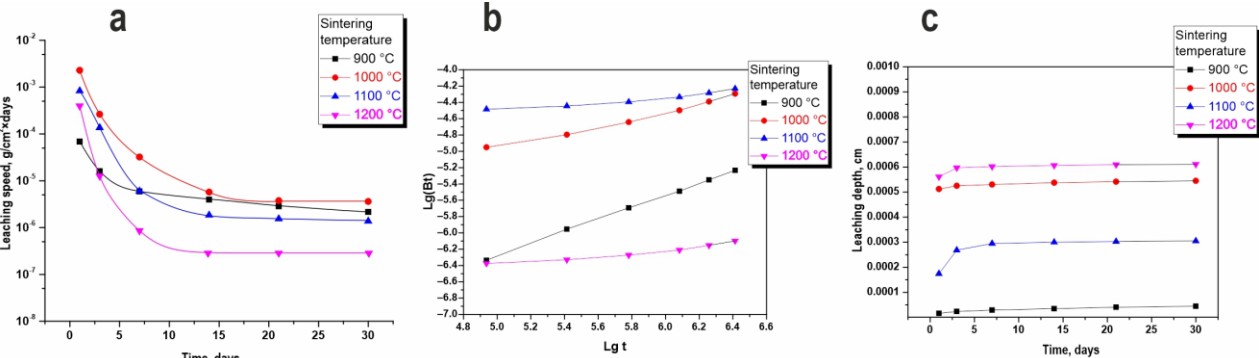

**Figure 7.** Hydrolytic stability of SrTiO$_3$ samples obtained via SPS with temperatures of 900, 1000, 1100, and 1200 °C: (**a**) leaching speed; (**b**) cumulative leaching fraction; (**c**) leaching depth.

All the samples were characterized by a change in the strontium elution mechanism during testing. For the 1000 °C sample, strontium was leached for the first 3 days via the mechanism of surface washing, after which the difference in chemical potentials—the driving force of the mass transfer process—became sufficient to compensate for the depletion of the surface layers by strontium diffusion from the sample bulk.

The 1100 and 1200 °C samples were characterized by a similar leaching mechanism as described above; however, they likely had an elevated strontium content at the surface. Additionally, the effective diffusion coefficient of strontium in these materials was lower. Therefore, the stage of compensating strontium leaching by its diffusion only occurred after 30 days. Furthermore, it seemed plausible that with increased test duration, a new change in mechanism could be recorded, with the limiting stage becoming the sample surface dissolution phase.

The leaching index (L) for all samples was above 8, allowing the conclusion that cesium is reliably immobilized within the material bulk and the synthesized matrices may be suitable for application as immobilizing materials [38].

Finally, to elucidate the leaching mechanism of Sr$^{2+}$ in SrTiO$_3$ matrices, we introduced two additional parameters—the diffusion coefficient and leaching index (L). Table 1 provides the diffusion coefficients of samples after processing at 900–1200 °C.

**Table 1.** Sr$^{2+}$ leaching parameters of SrTiO$_3$ samples obtained via SPS with temperatures of 900, 1000, 1100, and 1200 °C on the 30th day.

| Sintering Temperature, °C | D$_e$, cm$^2$/s | L | Depth, cm |
|---|---|---|---|
| 900 | $6.37 \times 10^{-11}$ | 10.20 | $4.48 \times 10^{-5}$ |
| 1000 | $6.02 \times 10^{-11}$ | 10.05 | $5.45 \times 10^{-5}$ |
| 1100 | $5.03 \times 10^{-11}$ | 10.30 | $3.05 \times 10^{-4}$ |
| 1200 | $1.73 \times 10^{-13}$ | 12.39 | $6.11 \times 10^{-4}$ |

As shown in Table 1, samples processed at temperatures below 1200 °C consisted of a dense monolithic solid structure, affecting their leachability parameters, which decreased

with increasing temperature. It should also be noted that penetration of the solution into ceramics was minimal and corresponded to the order of $10^{-4}$–$10^{-5}$. After processing at temperatures above 900 °C, all high-temperature products had a leachability index (L) above 9 and can be classified as allowing controlled disposal.

The calculated indicators of leaching depth are presented in Figure 7c. The ceramic samples had high stability in the medium of this solvent, due to their dense structure and the chemical stability of the matrices.

## 4. Conclusions

We conducted the synthesis of $SrTiO_3$ matrices using a sol–gel synthesis method according to Pechini and spark plasma sintering. The synthesized powder material was represented by a homogeneous phase composition of strontium titanate, obtained from precursor metallic salt precursors $Sr(NO_3)_2$ and $TiCl_3$. Solid-state matrices were obtained based on the synthesized powder by SPS in the temperature range of 900–1200 °C. Ceramics with a homogenous strontium distribution had high RD values (up to 99.79%) and micro-hardness (up to 950 HV). The sample obtained at 1200 °C showed the best performance of high hydrolytic stability, with a leaching rate of $10^{-7}$ g·cm$^{-2}$·day$^{-1}$ and a diffusion coefficient ($D_e$) of $1.73 \times 10^{-13}$ cm$^2$·s$^{-1}$.

The comparison of results allowed for the following conclusions to be drawn: Preparing the sol–gel starting mixture according to the method led to single-stage thermal compaction of the material, reducing the shrinkage rate and the temperature range of the process.

The ceramic formation mechanism is common in the case of reaction and normal sintering. Formation of the solid-state matrix occurred through the formation of aggregates of a dense monolithic structure that increased in size with increasing temperature. Formation of a fully developed ceramic compound was achieved at 1200 °C.

The obtained results can be applied to solve problems of producing perovskite-type ceramic minerals for immobilizing strontium radionuclides. The sol–gel synthesis technique can be used to reduce the stages of waste pre-treatment for compaction, particularly to eliminate the calcination stage.

**Author Contributions:** A.A.B.—writing—original draft, methodology, data curation, visualization, software; O.O.S.—conceptualization, writing—original draft; E.K.P.—conceptualization, project administration, writing—review and editing, resources; I.Y.B.—validation, writing—review and editing; E.S.K.—investigation, validation; O.V.K.—investigation, validation; S.A.A.—investigation, data curation; N.B.K.—investigation, validation; A.N.F.—investigation, validation; I.G.T.—conceptualization, supervising, writing—review and editing. All authors have read and agreed to the published version of the manuscript.

**Funding:** The study was financially supported within the State Assignment of the Ministry of Science and Higher Education of the Russian Federation, topic no. FZNS-2023-0003. The equipment of the joint center for collective use, the interdisciplinary center in the field of nanotechnology, and new functional materials of the FEFU were used in the work (Vladivostok, Russia).

**Data Availability Statement:** There are no databases or archives. All obtained results are displayed in the publication.

**Conflicts of Interest:** The authors declare that they have no known competing financial interests or personal relationship that could have appeared to influence the work reported in this paper.

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
