# Peer review of "Hybrid Sol–Gel and Spark Plasma Sintering to Produce Perovskite-like SrTiO3 Ceramics for Radioactive Waste Isolation"

_jcs, doi:10.3390/jcs7100421_

Round 1
Reviewer 1 Report
(a) The manuscript should be throughly revised in English.
(b) The "Introduction" section should be more concise, by shortening to less than one page.
(c) The panels in Fig. 2 can be combined.
(d) What will happen if the sintering temperature is further increased?
The authors are suggested to invite an editing agent to polish their manuscript.
Author Response
Re: « Combined sol-gel and spark plasma sintering to produce per-ovskite-like SrTiO3 ceramics» by Belov A.A. and et al. (Manuscript Number: jcs-2635593)
Dear dear Editors, dear Reviewers,
We deeply appreciate the time you spent reviewing our paper and the valuable recommendations you made. All the comments are taken into account and corresponding changes are made to the manuscript’s body text. Detailed point-by-point answers are presented below.
On behalf of co-authors,
Oleg Shichalin, Researcher, Ph.D.
Response to Reviewers
Reviewer #3:
Comment #1.
(a) The manuscript should be throughly revised in English.
Response to the Comment #1:
Thanks to the reviewer for the comment. Our paper has passed the peer review stage and the English language has been corrected by a native English speaker.
Comment #2.
The "Introduction" section should be more concise, by shortening to less than one page.
Response to the Comment #2:
Thanks to the reviewer for the comment. It is difficult to do so at the expense of elaboration, nuclear waste reprocessing is a very complex and responsible subject. There are no clear requirements for the length of the "introduction", please leave the text in its current form.
Comment #3.
The panels in Fig. 2 can be combined.
Response to the Comment #3:
Thanks to the author for the comment. This is an independent figure, SPS sintering processes are very important for specialists in the consolidation of highly dispersed powders
Comment #4.
What will happen if the sintering temperature is further increased?
Response to the Comment 4:
As the temperature rises further, melting of the material will occur. This process is not permissible for LRW disposal. Therefore, it is better not to bring the process to such a state.
Reviewer 2 Report
In this manuscript, Belov et al. reported the combined use of sol-gel with plasma sintering to produce perovskite-like SrTiO3 ceramics that can be applied for radioactive waste (RAW) isolation. Overall, the research topic is interesting and worthy of publication at J. Compos. Sci. The manuscript was also well drafted. However, to merit acceptance, some technical issues need to be resolved. Please see below for more details.
1. Please consider if the title needs some revision. The Introduction illustrates extensively the background information about radioactive waste (RAW) isolation. However, this was not reflected in the Title.
2. The Introduction needs some revision. (1) It overall appears to be a bit tedious. For instance, too much background information was given to RAW (the authors used two paragraphs). (2) Insufficient referencing was observed. For example, no references were given in the two paragraphs of RAW. References are also required in the paragraphs on page 3 (from line 105 to line 127).
3. When introducing perovskite oxides in page 2, related work can be referenced (e.g., Analytica Chimica Acta 2023, 1251, 341007).
4. Figure 1 needs some revision. (1) Figure 1a also included data of the sample STO 400. However, no discussion was given. If not relevant, this data might need to be deleted. (2) The bottom two figures of Figure 1b, the scale bars are the same for these two figures, however, the texture/morphology appears to be quite different. Why?
5. When discussing the sol-gel method in page 2, relevant works can be included (e.g., Small Methods, 2018, 2, 1800071).
6. Figure 4a*, 4b*, 4c*, and 4d*, for the inset figures, scale bars should be added.
7. Figure 7b, for the x axis and y axis, units should be provided.
Author Response
Re: « Combined sol-gel and spark plasma sintering to produce perovskite-like SrTiO3 ceramics» by Belov A.A. and et al. (Manuscript Number: jcs-2635593)
Dear dear Editors, dear Reviewers,
We deeply appreciate the time you spent reviewing our paper and the valuable recommendations you made. All the comments are taken into account and corresponding changes are made to the manuscript’s body text. Detailed point-by-point answers are presented below.
On behalf of co-authors,
Oleg Shichalin, Researcher, Ph.D.
Response to Reviewers
Reviewer #1:
In this manuscript, Belov et al. reported the combined use of sol-gel with plasma sintering to produce perovskite-like SrTiO3 ceramics that can be applied for radioactive waste (RAW) isolation. Overall, the research topic is interesting and worthy of publication at J. Compos. Sci. The manuscript was also well drafted. However, to merit acceptance, some technical issues need to be resolved. Please see below for more details.
Comment #1.
Please consider if the title needs some revision. The Introduction illustrates extensively the background information about radioactive waste (RAW) isolation. However, this was not reflected in the Title.
Response to the Comment #1:
Thanks to the reviewer for the comment. The title of the article was changed to "Hybrid sol-gel and spark plasma sintering to produce perov-skite-like SrTiO3 ceramics for radioactive waste isolation"
Comment #2.
The Introduction needs some revision. (1) It overall appears to be a bit tedious. For instance, too much background information was given to RAW (the authors used two paragraphs). (2) Insufficient referencing was observed. For example, no references were given in the two paragraphs of RAW. References are also required in the paragraphs on page 3 (from line 105 to line 127).
Response to the Comment #2:
Thanks to the author for the comment. The links have been added to the appropriate sections. The large amount of information devoted to RAW emphasizes the importance and relevance of the problem, which is being addressed by a large number of research teams.
Comment #3.
Response to the Comment #3:
Thanks to the reviewer for the comment. The relevant link has been added to the section
Comment #4.
Figure 1 needs some revision. (1) Figure 1a also included data of the sample STO 400. However, no discussion was given. If not relevant, this data might need to be deleted. (2) The bottom two figures of Figure 1b, the scale bars are the same for these two figures, however, the texture/morphology appears to be quite different. Why?
Response to the Comment #4:
Thanks to the reviewer for the comment. Unused information has been removed from the image. In Figure 1(b), the dimensional linearity is indeed the same for the two images. This is due to the fact that the left image shows a particle from the surface, while the right image shows another sol-gel particle lying on an edge. As a result, one can simultaneously see the surface of the particle and its internal organization, which also proves the agglomeration of nanoscale particles.
Comment #5.
When discussing the sol-gel method in page 2, relevant works can be included (e.g., Small Methods, 2018, 2, 1800071).
Response to the Comment #5:
Thanks to the reviewer for the comment. The link has been added to the appropriate section.
Comment #6.
Figure 4a*, 4b*, 4c*, and 4d*, for the inset figures, scale bars should be added.
Response to the Comment #6:
Thanks to the reviewer for the comment. The scale bar has been added to the corresponding figure
Comment #7.
Figure 7b, for the x axis and y axis, units should be provided.
Response to the Comment #7:
Thanks to the reviewer for the comment. Figure 7(b) shows an estimate of the dominant leaching mechanism resulting from logarithmic operations (formula 6 lines 215-218). This value has no dimensionality, therefore it was not indicated on the axes of Figure 7(b). In contrast to Figure 7(a), in which the leaching rate has a dimension, but the y-axis is logarithmic, in which case the dimension is specified. In Figure 7( b), the axes are standard.
Reviewer 3 Report
Comments for jcs-2635593:
1. In Figure 1a, the standard card information of SrTiO3 is suggested to put into for comparison.
2. In Page 2, line 74, The “ABO3” should be changed into “ABO3”.
3. In Page 6, line 239-241. The authors claim that they use the Rietveld refinement analysis, however, I did not see the Rietveld refinement image of SrTiO3 powder.
4. Based on the EDX results in Figure 5, the mole ratio between Sr and Ti is not close to 1:1. why?
5. In Page 7, line 272-275:
(1) Check the expression “as seen in Fig. 2a”;
(2) Check the units of cell parameters and add the volume cell of SrTiO3.
(3) When STO first appears, the corresponding full name should be clearly written.
6. In Table 1, use the “6.37×10-11” instead of “6.37·10-11”.

Author Response
Re: « Combined sol-gel and spark plasma sintering to produce perovskite-like SrTiO3 ceramics» by Belov A.A. and et al. (Manuscript Number: jcs-2635593)
Dear dear Editors, dear Reviewers,
We deeply appreciate the time you spent reviewing our paper and the valuable recommendations you made. All the comments are taken into account and corresponding changes are made to the manuscript’s body text. Detailed point-by-point answers are presented below.
On behalf of co-authors,
Oleg Shichalin, Researcher, Ph.D.
Response to Reviewers
Reviewer #2:
Comments for jcs-2635593:
Comment #1.
In Figure 1a, the standard card information of SrTiO3 is suggested to put into for comparison.
Response to the Comment #4:
Thanks to the reviewer for the comment. A card of the compound SrTiO3 is placed in the figure.
Comment #2.
In Page 2, line 74, The “ABO3” should be changed into “ABO3”.
Response to the Comment #2:
Thanks to the reviewer for the comment. Error corrected.
Comment #3.
In Page 6, line 239-241. The authors claim that they use the Rietveld refinement analysis, however, I did not see the Rietveld refinement image of SrTiO3 powder.
Response to the Comment #3:
Thanks to the reviewer for the comment. The powder for which the Rietveld thinning was carried out is shown in Figure 1. The Rietveld method is a theoretical refinement of the structure parameters, based on the calculation of data obtained from standard compounds from the database, with the data obtained by the XRD method. As a result of such processing, no new material is obtained, no physical manipulations are performed with it. Therefore, we cannot provide the requested image. Moreover, the software that allows obtaining data using the Rietveld method is not designed for visualization, only for obtaining numerical values.
Comment #4.
Based on the EDX results in Figure 5, the mole ratio between Sr and Ti is not close to 1:1. why?
Response to the Comment #4:
Thanks to the reviewer for the comment. It is unclear why the reviewer decided that the molar ratio of Sr:Ti should be 1:1. In Figure 5, the authors have not attached images describing the distribution and oxygen content, as the image would then be overloaded.
Comment #5.
In Page 7, line 272-275:
(1) Check the expression “as seen in Fig. 2a”;
(2) Check the units of cell parameters and add the volume cell of SrTiO3.
(3) When STO first appears, the corresponding full name should be clearly written.
Response to the Comment #5:
Thanks to the reviewer for the comment. All edits have been made
Comment #6
In Table 1, use the “6.37×10-11” instead of “6.37·10-11”.
Response to the Comment #6:
Thanks to the reviewer for the comment. Corrections made.
Round 2
Reviewer 1 Report
The manuscripy is recommended for publication after a careful check in English.
It is improved in English, only a minor revision is fine.
Reviewer 2 Report
The authors have addressed my comments and improved the manuscript accordingly. The revised manuscript is recommended for acceptance.
Author Response
Thank you for your assessment, our team is very grateful for your work.
Reviewer 3 Report
The authors have addressed the issues. It can be accepted now.
Author Response

(The authors gave the same response as above.)
